# The Efficacy of Polatuzumab Vedotin Targeting CD79B in the Treatment of Non-Hodgkin Lymphoma: A Systematic Review and Meta-Analysis

**DOI:** 10.3390/ijms26146836

**Published:** 2025-07-16

**Authors:** Samiyah Alshehri, Bushra Khan, Najeeb Ullah Khan, Ahsanullah Unar

**Affiliations:** 1Department of Pharmacology and Toxicology, College of Pharmacy, King Saud University, Riyadh 11421, Saudi Arabia; saalshehri@ksu.edu.sa; 2Institute of Biotechnology & Genetic Engineering (Health Division), The University of Agriculture Peshawar, Peshawar 25130, Pakistan; bushra27aup@gmail.com; 3Department of Precision Medicine, University of Campania ‘L. Vanvitelli’, 80138 Naples, Italy

**Keywords:** polatuzumab vedotin, non-Hodgkin lymphoma, CD79B, antibody-drug conjugate, overall survival

## Abstract

Polatuzumab vedotin (PoV) is a novel antibody-drug conjugate that targets CD79B for the treatment of Non-Hodgkin Lymphoma (NHL). This meta-analysis aimed to evaluate the efficacy and safety of PoV in patients with NHL. A systematic review and meta-analysis of clinical trials evaluating PoV in NHL were conducted. The primary outcomes were complete response (CR) rates, progression-free survival (PFS), and overall survival (OS). Safety outcomes were also assessed. Random-effects models were used for the pooled analyses. Thirteen studies with 1533 patients with NHL were included. PoV significantly improved CR rates compared to control treatments (OR 1.50, 95% CI 1.01–2.21, *p =* 0.04) and PFS (MD 4.17 months, 95% CI 2.18–6.15, *p* < 0.0001). OS was not significantly different (OR 0.97, 95% CI 0.47–2.01, *p* = 0.93). Adverse events were more common with PoV (RR 1.38, 95% CI 0.98–1.94, *p* < 0.0001). PoV improves CR rates and PFS in patients with NHL, particularly those with relapsed/refractory disease, but is associated with increased adverse events. Further research is needed on long-term survival outcomes and optimal patient selection. PoV appears to be a promising targeted therapy option for NHL, which warrants further investigation.

## 1. Introduction

Non-Hodgkin Lymphoma (NHL) is a diverse hematological malignancy that affects either T cells or B cells of the lymphatic system [1]. Approximately 4% of all cancer cases are diagnosed as NHL, which is a major cause of morbidity and mortality worldwide [2]. The most common subtypes of NHL are diffuse large B-cell lymphoma (DLBCL), follicular lymphoma (FL), and mantle cell lymphomas (MCL). Among all NHL cases, approximately 85% are commonly reported to have B-cell lymphomas, particularly DLBCL, which is an aggressive subtype [3]. Over the past few decades, the incidence of NHL has been progressively increasing in the aging population. The etiology of NHL is multifactorial and includes genetic, immunological, and environmental factors [4]. Despite increasing advancements in therapeutic strategies for NHL patients, relapsed or refractory patients exhibit poor outcomes owing to resistance to traditional therapeutic strategies such as chemo-immunotherapy [5,6]. Thus, targeted therapies have become a significant research area, as they directly target the specific molecular pathways of lymphomagenesis to improve patient outcomes [7].

Among these targeted therapies, polatuzumab vedotin (Polivy^®^) has emerged as a novel antibody–drug conjugate (ADC), having a humanized IgG1 antibody that targets the CD79b gene, which is an important signaling factor of B-cell receptors [8]. CD79B is also conjugated to a potent cytotoxin known as monomethyl auristatin E (MMAE), which aids in the development of an anticancer microenvironment after the binding of polatuzumab vedotin to CD79b in B-cell malignancies [8,9]. The internalization of polatuzumab vedotin causes the intracellular release of MMAE, which interferes with microtubule assembly and ultimately results in cell death. Early clinical trials have shown that this specific method of action is effective in treating patients with challenging-to-treat lymphomas [10]. Additionally, B-cell receptor pathways play a significant role in the proliferation and survival of B cells, making it a crucial target for therapeutic interventions in B-cell lymphoma [11].

In various NHL subtypes, the rapid activation of these pathways is a major biomarker because of their role in tumor progression and survival. Polatuzumab vedotin is a targeted therapy that directly controls this pathway by targeting and eradicating malignant B cells, without damaging healthy tissues. Thus, this targeted drug reduces the risks associated with off-target toxicity, as reported in conventional chemotherapies [12,13].

Recently, the emergence of polatuzumab vedotin as a targeted therapy has altered the mode of therapeutic management of relapsed or refractory DLBCL and sparked research interest in the oncology community [14]. This approach reduces the systemic toxicities frequently seen with conventional chemotherapeutic drugs by minimizing the effects on healthy tissues and non-malignant B cells [15]. With polatuzumab vedotin demonstrating evident benefits in patients with R/R DLBCL who had received extensive pretreatment, encouraging preclinical and early clinical findings have validated CD79B as a viable target [13].

A growing body of evidence supports the efficacy of polatuzumab vedotin in NHL, particularly R/R DLBCL. Previous clinical trials have reported the potential outcomes of polatuzumab vedotin among NHL patients [16,17]. However, no comprehensive study has evaluated the efficacy and safety of this therapy. Thus, this study aimed to evaluate the efficacy of polatuzumab vedotin as a CD79B gene-targeting drug across different subgroups of Non-Hodgkin Lymphoma patients by adopting a systematic review and meta-analysis research approach.

## 2. Methods

### 2.1. Search Design

The “Reporting Items for Systematic Review and Meta-Analysis (PRISMA)” guidelines [18] were applied to perform this meta-analysis evaluating the efficacy of polatuzumab vedotin (Pola) as a CD79B gene-targeting drug for the treatment of Non-Hodgkin Lymphoma. This study was a pooled analysis of previously published randomized controlled trials (RCTs); therefore, no additional ethical review was required.

### 2.2. Compliance with Registration

This study adhered to the PRISMA guidelines for systematic reviews and meta-analyses. No formal protocol was registered in a public database such as PROSPERO. However, the methods were defined a priori and rigorously followed throughout the study.

#### 2.2.1. PICO Framework

This study used the Population Intervention Control Outcome and Study (PICOS) framework [19] to guide the search.

Population: Patients (>18 years) diagnosed with Non-Hodgkin Lymphoma, particularly those with B-cell subtypes, such as diffuse large B-cell lymphoma (DLBCL) and follicular lymphoma (FL).

Intervention: The use of polatuzumab vedotin (Pola) alone and in combination with other chemotherapeutic drugs such as bendamustine or rituximab.

Comparison: CAR- T therapy or other drugs (bendamustine and rituximab) or placebo.

Outcomes: The patients’ outcomes were complete response (CR), overall survival (OS), progression-free survival (PFS), and. Secondary outcomes were related to safety (adverse events 3–5).

Study Design: Randomized controlled trials (RCTs).

#### 2.2.2. Search Strategy

The PRISMA guidelines assisted in the selection of research articles related to the study aims. Three electronic databases, PubMed, EMBASE, and ClinicalTrials. gov, were searched from inception to December 2024. The MeSH keywords used for searching research articles from databases were [(“Polatuzumab Vedotin” OR “Polivy”) AND (“CD79B” OR “B-cell receptor” OR “B-cell receptor signaling” OR “CD79B targeting”) AND (“Efficacy” OR “Disease-free survival” OR “Overall survival” OR “Progression-free survival” OR “Complete response rate”) AND (“Non-Hodgkin Lymphoma” OR “NHL” OR “Diffuse large B-cell lymphoma” OR “DLBCL” OR “B-cell lymphoma” OR “Mantle cell lymphoma” OR “Follicular lymphoma”). We carefully examined the reference lists of all previous systematic reviews and meta-analysis-based articles to search for further research articles.

### 2.3. Eligibility Criteria

The eligibility criteria were used to select and screen research articles after searching for research articles from electronic databases.

#### Inclusion Criteria

Studies that analyzed patients (>18 years) diagnosed with Non-Hodgkin Lymphoma (NHL).Studies have analyzed the use of polatuzumab vedotin (Pola) alone and in combination with other chemotherapeutic drugs (such as bendamustine or rituximab).Studies tracking patients’ outcomes related to complete remission (CR), overall survival rates (OS), progression-free survival (PFS), and safety outcomes (adverse events 3–4).Primary research studies such as cohort studies, randomized controlled trials (RCTs).Studies with full text available and published in English language.

### 2.4. Exclusion Criteria

The following studies were excluded:Studies analyzing patients without Non-Hodgkin Lymphoma (NHL).Studies not related to the effects of polatuzumab vedotin.Studies analyzing outcomes other than those mentioned above.Studies based on systematic reviews, meta-analyses, comprehensive reviews, narrative reviews, and editorials.Studies published in languages other than English and non-full-text papers.Studies with insufficient data or incomplete methodology.

### 2.5. Data Extraction

Two independent reviewers extracted the data to be placed in a prespecified table. Data related to demographic information, such as authors, year of study, country, study population, study design, treatment dose, study groups, and efficacy outcomes, were extracted (Table 1). The safety-related outcomes of the included studies are listed in Table 2. Discrepancies were resolved by consulting with a third reviewer.

### 2.6. Risk Bias Assessment

The Cochrane risk-of-bias assessment tool was used to evaluate the risk of bias in the included RCTs [20]. Bias was assessed based on seven domains: (a) allocation concealment, (b) selection bias or random sequence generation, (c) performance bias or the blinding of participants and personnel, (d) detection bias or blinding of outcome assessment, (e) selective bias or selective reporting, and other biases. Each domain’s score was categorized as low-risk, high-risk, or unclear.

### 2.7. Statistical Analysis

Review Manager Software (Cochrane Collaboration, version 5.4.0) was used to conduct all statistical analyses. A pooled analysis of data was performed for studies with potential heterogeneity using random-effects models. Statistical significance was set at *p* < 0.05 was considered statistically significant [21]. Effect sizes are shown as the mean difference for continuous outcomes such as progression-free survival (PFS) and overall survival (OS). The odds ratio was calculated using the number of events, such as complete response and overall survival rates. Heterogeneity was evaluated using the I^2^ statistic, with I^2^ values > 50% indicating significant heterogeneity.

**Table 1 ijms-26-06836-t001:** Characteristics of included studies.

Author, Year	Country	Study Design	Study Population (Median Age)	Study Follow Up	Dose of Drugs	Study Groups	Complete Response Rates	Overall Survival (OS)	PFS
Hatake et al., 2016 [22]	Japan	Phase I study	7 patients B-NHL (62 years) FL: 4 DLBCL: 3	3 months	1.0 or 1.8 mg/kg of PoV	DLCBL: 3 patients FL: 4 patients	G1: 2 G2: 1		
Terui et al., 2021 [23]	Japan	Phase I study	35 DLBCL patients (71 years)	5.4 months	1.8 mg/kg intravenously (IV) of PoV + bendamustine 90 mg/m^2^ IV + rituximab 375 mg/m^2^ IV (Pola + R-CHP)		G1: 12 G2: 11		G1: 5.2 months G2: 3.6 months
Tilly et al., 2021 [24]	France	Phase 3 RCT trial	879 DLBCL patients (49 years)	28.2 months	G1: 1.8 mg/kg IV PoV (pola-R-CHP) G2: 1.4 mg/m^3^ + 375 mg of rituximab + 750 mg cyclophosphamide + doxorubicin (R-CHOP)	pola-R-CHP: 440 R-CHOP: 439	G1: 381 G2: 363	G1: 53 G1: 57	
Wang et al., 2023 [25]	USA	Phase Ib/II study	20 MCL patients (68 years)	15.8 months	G1: Mosunetuzumab + polatuzumab vedotin (1.8 mg/kg intravenous [IV] infusion) G2: Bruton’s tyrosine kinase inhibitor (BTKi) therapy	M + PoV: 20 BTKi: 20	G1: 14 G2: 11	G1: 14 G2: 17	
Budde et al., 2021 [26]	USA	Phase Ib/II study	22 B-NHL patients (70 years)	9.6 months	G1: Mosunetuzumab + polatuzumab vedotin (1.8 mg/kg intravenous [IV] infusion) G2: CAR-T therapy	M + PoV: 22 Previous CAR-T therapy: 10	G1: 12 G2: 5		
Morschhauser et al., 2019 [27]	France	Phase 2 study	81 DLBCL (69 years) patients	1 year	G1: rituximab + polatuzumab vedotin (1.8 mg/kg intravenous [IV] infusion) G2: 375 mg/m^2^ rituximab plus 2.4 mg/kg ADCs	R-pola: 42 R-Pina: 39	G1: 8 G2: 11		
Morschhauser et al., 2014 [28]	France	Phase 2 study	122 DLBCL patients (60 years)	10 months	G1: polatuzumab vedotin (1.8 mg/kg) + rituximab G2; pinatuzumab vedotin + rituximab	PoV + R: 59 PiV + R: 63	G1: 6 G2: 10	G1: 22 G2: 24	G1: 5.4 mo G2: 5.2 mo
Philips et al., 2016 [29]	USA	Phase Ib/II study	70 patients (71 years)	3 months	1.8 mg/kg intravenously (IV) of PoV	DLBCL: 38 FL: 32	G1: 6 G2: 7	G1: 11 G2: 18	G1: 11.5 G2: 2.8
Philips et al., 2021 [30]	USA	Phase 1b/2 study	88 patients (65 years)	10.7 months	1.8 mg/kg of PoV + obinutuzumab	DLBCL: 45 FL: 43	G1: 13 G2: 0		
Bartlett et al., 2015 [31]	USA	Phase 1b/2 study	13 patients (68 years)	10.9 months	1.8 mg/kg of PoV + rituximab, cyclophosphamide, doxorubicin, and prednisone	DLCBL: 10 FL: 3	G1: 5 G2: 1		
Tilly et al., 2016 [32]	France	Phase Ib/II Study	36 DLBCL patients (70 years)	11.8 months	Polatuzumab vedotin combined with rituximab (Pola-R-CHP), cyclophosphamide, doxorubicin, and prednisone (R-CHP)	Pola-R-CHP: 26 R-CHP: 10	G1: 20 G2: 3		
Sehn et al., 2017 [33]	Canada	Phase Ib/II Study	80 R/R DLBCL (71 years)	10.9 months	G1: 1.8 mg/kg of PoV + bendamustine (B) + rituximab (375 mg/m^2^) G2: 1.8 mg/kg of PoV + bendamustine (B) + rituximab (375 mg/m^2^) obinutuzumab	Pola + BR: 39 Pola + BG: 39	G1: 23 G2: 8	G1: 20 G2: 8	G1: 6.7 G2: 2.0
Sehn et al., 2018 [34]	Canada	Phase Ib/II Study	80 R/R DLBCL (71 years)	30 months	G1: 1.8 mg/kg of PoV + bendamustine (B) + rituximab (375 mg/m^2^) G2: 1.8 mg/kg of PoV + bendamustine (B) + obinutuzumab	Pola + BR: 39 Pola + BG: 39	G1: 7 G2: 6		G1: 7.6 G2: 2.0

PoV, polatuzumab vedotin; MCL, mantle cell lymphoma; FL, follicular lymphoma; DLBCL, diffuse large B-cell lymphoma; BR, bendamustine + rituximab, BG, bendamustine (B) + obinutuzumab (G), B-NHL: B-cell non-Hodgkin lymphoma, R/R: refractory/relapsed, and PFS: progression-free survival.

**Table 2 ijms-26-06836-t002:** Safety outcomes of treatment among NHL patients.

Author, Year	Safety Outcomes	Neutropenia	Fatigue	Diarrhea	Nausea
Hatake et al., 2016 [22]	AE > 3: 4 AE > 4: 1	12		9	12
Terui et al., 2021 [23]	AE > 3: 31 AE > 4: 12				
Tilly et al., 2021 [24]	AE > 3: 230 AE < 4: 7	G1: 134 G2: 143	G1: 112 G2: 116	G1: 134 G2: 88	G1: 181 G2: 161
Wang et al., 2023 [25]	AE > 3: 10 AE < 4: 2		9	6	5
Budde et al., 2021 [26]	AE > 3: 11 AE < 4: 2	8	8	8	8
Morschhauser et al., 2019 [27]	G1: 30 G2: 33	G1: 9 G2: 12		G1: 4 G2: 3	
Morschhauser et al., 2014 [28]	G1; 21 G2: 27 AE > 3: 30 AE < 4: 9	G1: 27 G2: 17	G1: 32 G2: 2	G1: 25 G2: 4	
Philips et al., 2016 [29]	AE > 3: 30 AE < 4: 20	G1: 2 G2: 9			
Philips et al., 2021 [30]	AE > 3: 48 AE > 4:	18	23	16	
Bartlett et al., 2015 [31]	AE > 3: 10 AE < 4: 6	5	7	6	8
Tilly et al., 2016 [32]	AE > 3: 16 AE < 4: 12	8	2	10	10
Sehn et al., 2017 [33]	G1: 5 G2: 7	G1; 21 G2: 15			
Sehn et al., 2018 [34]					

G1: group 1, G2: group 2, AE: adverse events.

## 3. Results

### 3.1. Search Results

The selection and screening of research articles related to the study aim “Efficacy of polatuzumab vedotin as CD79B gene targeting drug for treatment of Non-Hodgkin Lymphoma patients” was performed according to the PRISMA guidelines in this systematic review and meta-analysis. A total of 1020 research articles were extracted after applying the aforementioned search strategy. Only 610 papers were initially screened, and 187 research articles were retrieved before the final screening. Among these, only 60 articles were assessed for eligibility criteria, and the final number of research articles after applying the exclusion criteria was 8, as shown in Figure 1.

### 3.2. Characteristics of Included Studies

Our study analyzed 13 studies (Phase I–III) and 1533 Non-Hodgkin Lymphoma (NHL) patients to evaluate the effectiveness of polatuzumab vedotin as a CD79B gene-targeting drug using a meta-analysis approach. The main characteristics of the selected studies for the pooled analysis are presented in Table 1 and Table 2. All included studies were phase I, II, and III studies published between 2000 and 2025. The sample size for the included studies ranged from 7 to 879 NHL patients (subtypes R/R DLBCL, FL, and MCL) from included studies. The median age of the patients was 69 years. The follow-up period of the included clinical trials ranged from 3 months to 30 months. All included studies employed polatuzumab vedotin (1.8 mg/kg intravenously (IV)) alone or with other chemotherapeutic drugs, namely bendamustine (90 mg/m^2^ IV) and rituximab (375 mg/m^2^ IV), in comparison to chemotherapeutic drugs alone or pinatuzumab vedotin.

### 3.3. Findings from Risk of Bias Assessment

The Cochrane risk tool was used for risk bias assessment of all included studies owing to their RCT design. All the included studies were at low risk, as shown in Figure 2 and Figure 3.

### 3.4. Primary Outcomes

#### 3.4.1. Complete Response Rates (CR)

All included studies reported complete response (CR) rates as an outcome of polatuzumab vedotin among Non-Hodgkin Lymphoma (NHL) patients. The pooled analysis showed that rates of CR significantly improved among NHL patients after PoV compared to placebo [OR: 1.50 CI 95%: 1.01–2.21; *p* = 0.04; I^2^ = 48%]. The subgroup analysis was performed for two subgroups: PoV versus placebo, showing higher CR rates among the group receiving PoV as compared to placebo [OR: 1.41 (CI 95%: 0.94–2.11)] and DLBCL versus FL showing higher CR rates among DLBCL patients as compared to FL [OR: 3.42 (0.57–20.33)], as shown in Figure 4. The asymmetrical distribution of the studies showed a high publication bias among the included studies, as shown in Figure 5.

#### 3.4.2. Progression Free Survival (PFS)

Five included studies reported progression-free survival (PFS) rates as an outcome of polatuzumab vedotin among patients with Non-Hodgkin Lymphoma (NHL). The pooled analysis showed that PFS in months was significantly improved in the experimental group using PoV compared to placebo [MD: 4.17 CI 95%: 2.18–6.15; *p* < 0.0001, I^2^ = 99%], as shown in Figure 6. The slightly symmetrical distribution of studies showed moderate publication bias among the included studies, as shown in Figure 7.

#### 3.4.3. Overall Survival Rates (OS)

Five of the included studies reported complete response (OS) rates as an outcome of polatuzumab vedotin among Non-Hodgkin Lymphoma (NHL) patients. The pooled analysis showed that rates of OS were not significantly associated with the use of PoV as compared to placebo (OR: 0.97 (CI 95%: 0.47–2.01) *p* = 0.93, I^2^= 73%), as shown in Figure 8. The asymmetrical distribution of studies showed a high publication bias among the included studies, as shown in Figure 9.

#### 3.4.4. Safety Outcomes

Among the 13 included studies, 5 reported adverse events such as neutropenia, fatigue, and diarrhea as adverse events after treatment with PoV as compared to placebo. The pooled analysis showed that adverse events were higher in the experimental group (using PoV) than in the placebo group [RR: 1.38 (0.98 to 1.94) *p* < 0.0001, I^2^ = 82%], as shown in Figure 10.

## 4. Discussion

This study aimed to evaluate the effectiveness of polatuzumab vedotin as a CD79B gene-targeting drug for the treatment of Non-Hodgkin Lymphoma (NHL) by adopting a systematic review and meta-analysis research approach. Through an analysis of 13 studies (Phase I-III) and 1533 Non-Hodgkin Lymphoma (NHL) patients, the findings of the study reported that complete response (CR) rates and progression-free survival (PFS) (in months) increased in the experimental group receiving polatuzumab vedotin as compared to the control group (chemotherapeutic drugs), while the rates of adverse events (such as neutropenia, diarrhea, and fatigue) were higher in the experimental group than in the placebo group. The pooled analysis showed that rates of CR [OR: 1.50 CI 95%: 1.01–2.21], *p* = 0.04, I^2^ = 48%] and PFS (in months) [MD: 4.17 CI 95%: 2.18–6.15, *p* < 0.0001, I^2^ = 99%] significantly improved among NHL patients after PoV as compared to placebo. Contrastingly, the pooled analysis also reported that adverse events were higher in the experimental group (using PoV) than in the placebo group [RR: 1.38 (0.98 to 1.94) *p* < 0.0001, I^2^ = 82%]. Furthermore, overall survival rates (OS) were not significantly associated with PoV, and the reason for that may be the lack of available data on OS. The slight symmetrical to asymmetrical distribution of studies on funnel plots showed low-to-moderate publication bias among the included studies. The heterogeneity ranged from 48% to 99% among the included studies owing to the involvement of NHL subtypes, different study designs, and study populations. All included studies were low risk, as assessed using the Cochrane risk tool.

These findings were consistent with previous clinical trials that reported effective clinical outcomes after the use of PoV as compared to chemotherapeutic drugs among NHL patients (e.g., DLBCL and FL). The potential of PoV to target CD79B, a molecule essential for B-cell receptor signaling and a major player in the pathophysiology of B-cell malignancies such as NHL, has been repeatedly investigated in prior research. An antibody-drug combination (ADC) called polatuzumab vedotin limits harm to healthy cells by directly delivering a strong cytotoxic chemical to cancer cells that express CD79B. Patients treated with PoV showed encouraging CR rates and PFS-related outcomes in a number of clinical trials, including critical phase II studies. This is especially true for patients with refractory or recurrent diffuse large B-cell lymphoma (DLBCL), a subtype of NHL. Sehn et al., 2017 [33] reported that PoV was associated with better CR rates, as shown in the current meta-analysis. The combination of PoV with bendamustine and rituximab (BR) resulted in noticeably higher CR rates (40%) than the BR regimen alone (18%). The PoV combination also prolonged PFS (12.4 months vs. 7.6 months), which supports the current study’s finding that PoV improves progression-free survival.

Moreover, the mechanistic rationale for using polatuzumab vedotin in NHL is supported by preclinical evidence demonstrating its selective affinity for CD79B, which is ubiquitously expressed on malignant B cells but limited in normal tissues. This targeted approach not only enhances therapeutic efficacy but also minimizes systemic toxicity compared to conventional chemotherapy. Importantly, the antibody-drug conjugate format of PoV, which links a monoclonal antibody to monomethyl auristatin E (MMAE), facilitates efficient intracellular delivery of the cytotoxic agent following CD79B-mediated internalization. This mode of action underscores the observed improvements in complete response and PFS seen in this meta-analysis. These findings support the integration of PoV into combination regimens as a rational strategy to optimize outcomes in relapsed or refractory NHL, particularly among patient’s ineligible for high-dose chemotherapy or stem-cell transplant. As treatment paradigms continue to shift toward biomarker-guided therapies, PoV represents a critical step toward personalized medicine in hematologic oncology.

However, previous experiments have revealed inconsistent outcomes when comparing OS. While some studies showed a trend towards better OS with PoV, others did not find any statistically significant improvements. This could be due to differences in the follow-up periods, previous therapies, and patient characteristics. The lack of a meaningful correlation between PoV and OS in the current meta-analysis may be due to the heterogeneity of patient populations in the included studies or the restricted availability of data. Specifically, the heterogeneity in patient demographics (e.g., age, comorbidities), treatment histories (such as number and type of prior therapies), and biological differences among NHL subtypes (e.g., variations in tumor aggressiveness, molecular profiles, and responsiveness to antibody-drug conjugates like polatuzumab vedotin) may have contributed to the modest impact on OS. In several included studies, the benefits in progression-free survival or response rates may not have translated into improved OS due to subsequent lines of therapy or crossover effects. Additionally, the current data do not allow for a definitive conclusion regarding first-line use, as only a limited number of studies (such as the POLARIX trial) have evaluated polatuzumab vedotin in previously untreated patients.

The use of liquid biopsy techniques is a significant future trend in the treatment of lymphoma, especially when using targeted medicines like polatuzumab vedotin. The non-invasive detection of circulating tumor DNA (ctDNA) made possible by liquid biopsy offers real-time information on tumor burden, clonal evolution, and response to treatment. The potential of ctDNA monitoring to distinguish early responders from non-responders has been shown by recent research, such as the study by Almasri et al. [35], which can help guide therapeutic decisions and reduce needless exposure to unsuccessful treatments. Liquid biopsy has the potential to greatly improve response classification and enable more individualized treatment strategies in polatuzumab vedotin trials and real-world investigations.

However, this study had a few limitations with enormous strengths. First, the number of included studies was limited, which compromised the heterogeneity and sensitivity of these studies. Second, the inclusion of single-arm studies reduced the validity of the findings of this meta-analysis. Third, the limited number of participants reduced the authenticity of the findings in this meta-analysis. Larger RCTs should be conducted to validate the clinical implications of PoV among patients with NHL. The findings of this study emphasized the need for future research with broader age representation and standardized adverse event reporting to better assess these variables.

## 5. Conclusions

Overall, the findings of this meta-analysis reported that olatuzumab vedotin is effective in increasing CR rates and PFS in NHL patients, especially those with relapsed or refractory disease. However, the rate of adverse events was higher among NHL patients receiving PoV than among those receiving placebo. Nonetheless, further research is necessary, given the elevated risk of adverse events and lack of a meaningful correlation with OS. Subgroup analyses of various NHL subtypes, more standardized adverse event reporting, and long-term follow-up data on overall survival are required to maximize the utility of PoV in clinical practice. Although PoV is a promising therapeutic alternative, its clinical value must be maximized by cautious patient selection and adverse event management.

## Figures and Tables

**Figure 1 ijms-26-06836-f001:**
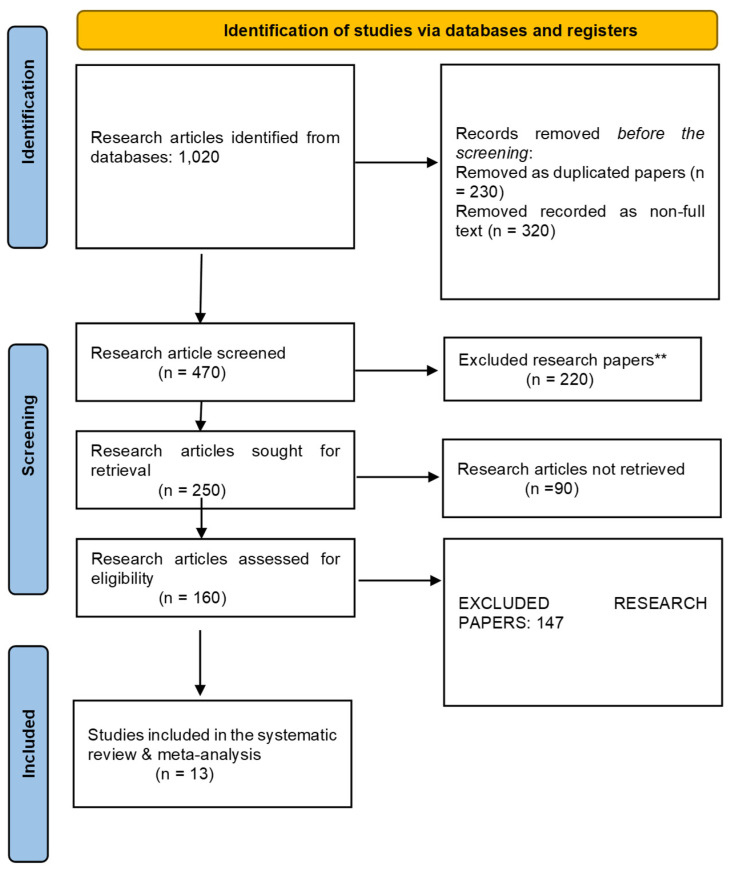
The PRISMA flowchart for the screening and selection of included studies. ** means *p* < 0.01.

**Figure 2 ijms-26-06836-f002:**
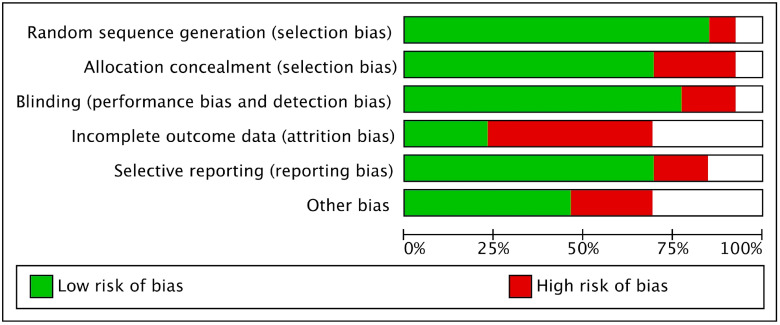
A graph representing risk of bias in the included studies.

**Figure 3 ijms-26-06836-f003:**
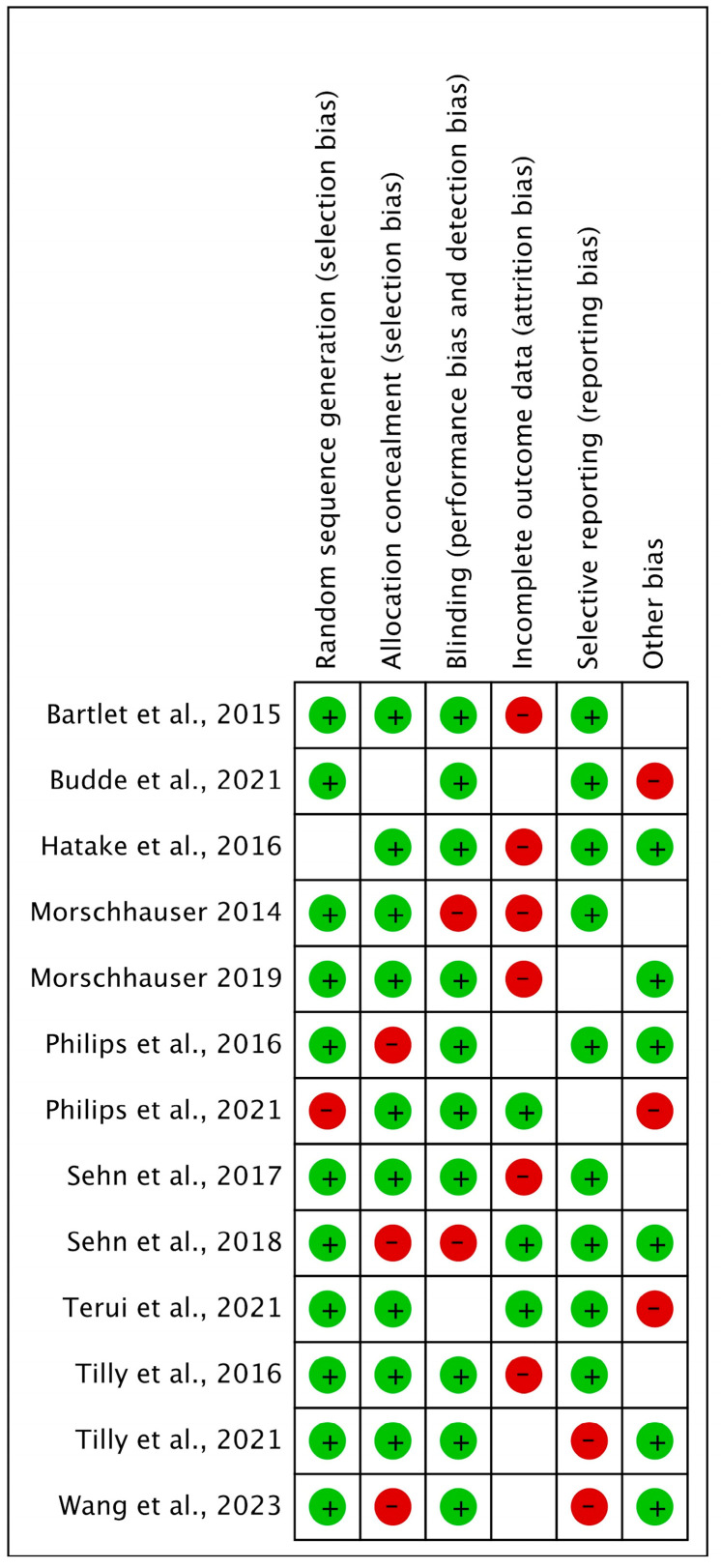
A summary of the included studies representing the risk of bias [22,23,24,25,26,27,28,29,30,31,32,33,34].

**Figure 4 ijms-26-06836-f004:**
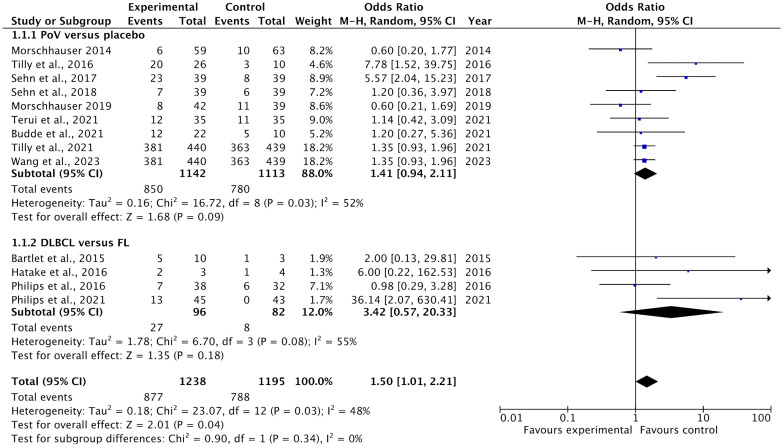
A forest plot of the odds ratio of CR rates among the experimental and control groups [22,23,24,25,26,27,28,29,30,31,32,33,34].

**Figure 5 ijms-26-06836-f005:**
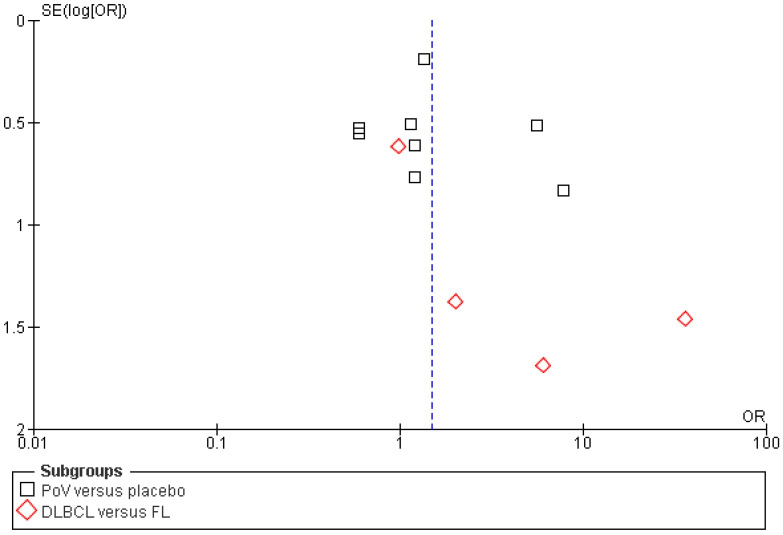
Funnel plot of the odds ratio (OR) for complete response (CR) rates between experimental and control groups. Each data point represents an individual study plotted against its standard error [SE(logOR)]. Black squares indicate studies comparing PoV versus placebo, while red diamonds indicate comparisons between DLBCL and FL subgroups. The vertical blue dashed line represents the pooled odds ratio estimate from the meta-analysis.

**Figure 6 ijms-26-06836-f006:**
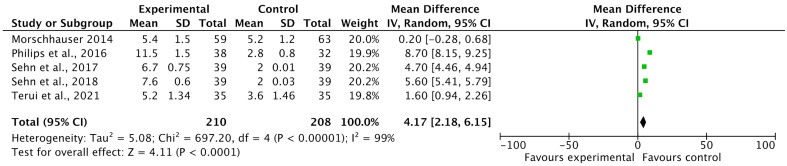
A forest plot of the mean difference of PFS (in months) among the experimental and control groups [23,28,29,33,34].

**Figure 7 ijms-26-06836-f007:**
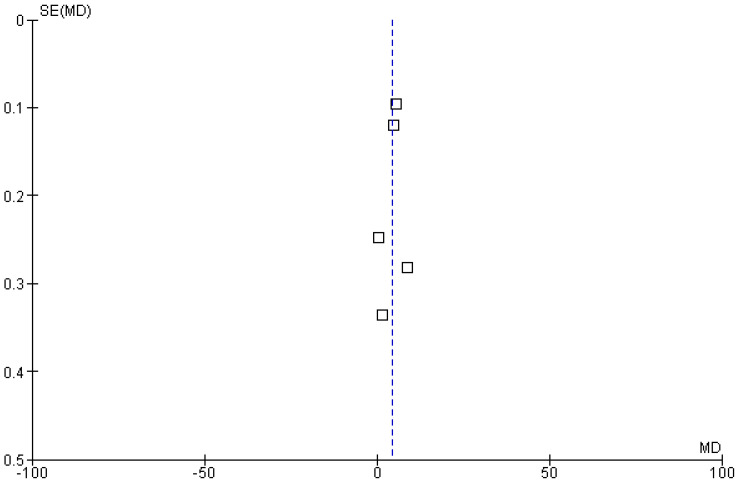
Funnel plot of the mean difference in progression-free survival (PFS, in months) between experimental and control groups. Each square represents an individual study plotted against its standard error. The vertical blue dashed line indicates the pooled effect estimate (overall mean difference) from the meta-analysis.

**Figure 8 ijms-26-06836-f008:**
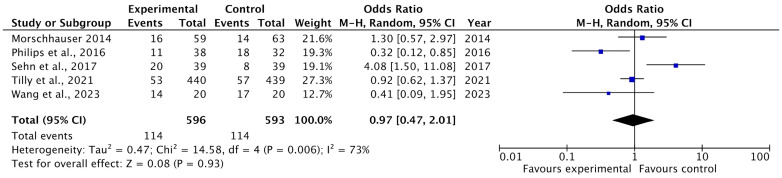
A forest plot of the odds ratio of OS rates among the experimental and control groups [24,25,28,29,33].

**Figure 9 ijms-26-06836-f009:**
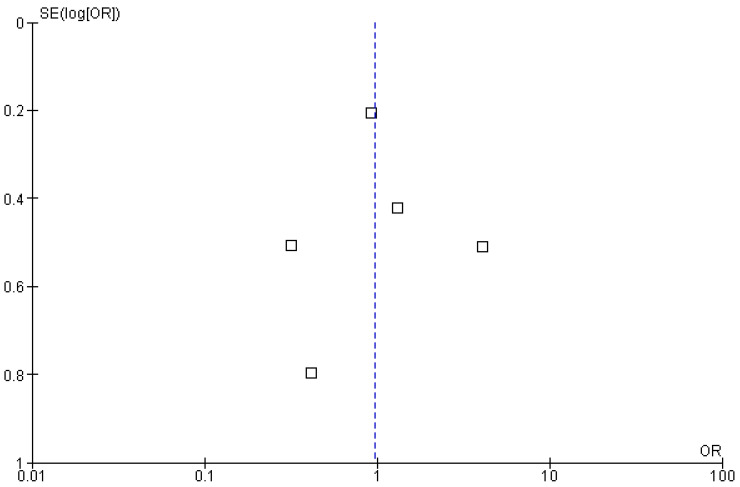
Funnel plot of the odds ratio (OR) for overall survival (OS) rates between experimental and control groups. Each square represents an individual study plotted against its standard error [SE(logOR)]. The vertical blue dashed line indicates the pooled odds ratio estimate derived from the meta-analysis.

**Figure 10 ijms-26-06836-f010:**
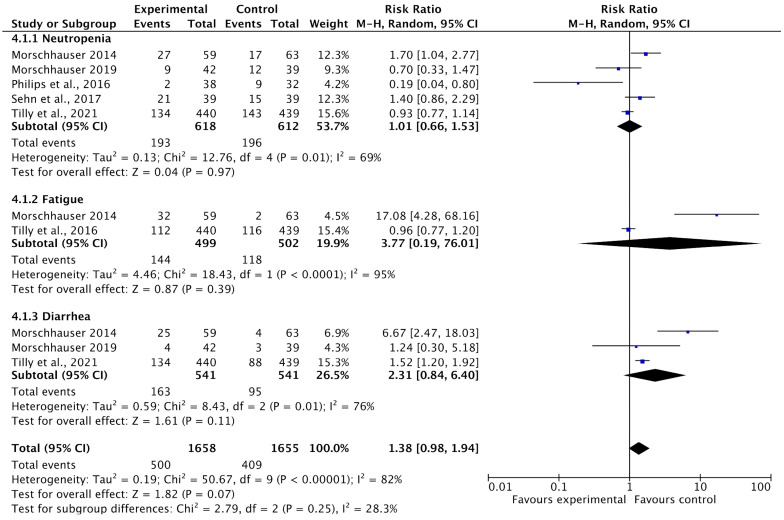
A forest plot of the risk ratio of adverse events among the experimental and control groups [24,27,28,29,32,33].

## Data Availability

The manuscript includes all necessary data, and related data may be provided upon request from the corresponding authors.

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
