# Peer review of "The Efficacy of Polatuzumab Vedotin Targeting CD79B in the Treatment of Non-Hodgkin Lymphoma: A Systematic Review and Meta-Analysis"

_ijms, 2025, doi:10.3390/ijms26146836_

Round 1
Reviewer 1 Report
Comments and Suggestions for Authors
In this study, Alshehri et al. report the results of a systematic review and meta-analysis of thirteen studies assessing polatuzumab vedotin and including more than 1300 patients with non-Hodgklin lymphoma. The results document that use of polatuzumab vedotin for the treatment of B cell non-Hodgkin lymphoma improves the rate of complete response and of progression free survival, though not overall survival. The data also document that polatuzumab vedotin associates to a higher frequency of adverse events compared to standard therapy. The study is well performed and is clinically relevant and meaningful in the current therapeutic scenario of B cell non-Hodgkin lymphoma treatment. A few issues require to be tackled for further improvement, as detailed in the comments below.
MAJOR ISSUES
- The authors should clarify whether the same conclusions apply to use of polatuzumab vedotin both in first line treatment and in relapsed/refractory non-Hodgkin lymhoma. Were there any differences?
- On the same line, table 1 should detail more precisely the clinical context of the trials (first line? Relapsed/refractory?)
- The last sentence of page 5 “All included studies employed…” may be somehow confusing and should be reformulated. Is “pinatuzumab vedotin” correct?
- Could the authors define whether the rate of adverse events listed in their systematic review was dependent upon age? Or upon the specific treatment regimen that was utilized?
- In view of Precision Medicine, one important issue is trying to understand which patients benefit the most from polatuzumab vedotin based on the molecular characteristics of the disease. Importantly, liquid biopsy of circulating tumor DNA has been shown to identify patients with optimal responses to standard treatment with R-CHOP, as recently reviewed and updated in detail (Almasri et al., Int J Mol Sci. 2025, 19;26(10):4869. doi: 10.3390/ijms26104869). In their Discussion, the authors should mention the role of liquid biopsy in lymphoma and state that an important future perspective will be to apply liquid biopsy to studies using polatuzumab vedotin in order to facilitate the early identification of responders versus non-responders (e.g. Moia et al. Blood Adv. 2025, 8;9(7):1692-1701. doi: 10.1182/bloodadvances.2024014136).
MINOR ISSUES
- In the Introduction, the authors correctly state that the etiology of non-Hodgkin lymphoma is multifactorial. It would be appropriate to state also that the etiology is unknown for most cases, whereas the molecular pathogenesis of these neoplasms is rather well established.
Author Response
R1
Comments and Suggestions for Authors
In this study, Alshehri et al. report the results of a systematic review and meta-analysis of thirteen studies assessing polatuzumab vedotin and including more than 1300 patients with non-Hodgklin lymphoma. The results document that use of polatuzumab vedotin for the treatment of B cell non-Hodgkin lymphoma improves the rate of complete response and of progression free survival, though not overall survival. The data also document that polatuzumab vedotin associates to a higher frequency of adverse events compared to standard therapy. The study is well performed and is clinically relevant and meaningful in the current therapeutic scenario of B cell non-Hodgkin lymphoma treatment. A few issues require to be tackled for further improvement, as detailed in the comments below.
Major issues
- The authors should clarify whether the same conclusions apply to use of polatuzumab vedotin both in first line treatment and in relapsed/refractory non-Hodgkin lymhoma. Were there any differences
Response: Dear Reviewer, thank you for your insightful comment. We acknowledge the importance of distinguishing between the use of polatuzumab vedotin in first-line versus relapsed/refractory (R/R) settings in non-Hodgkin lymphoma (NHL). In our study, we primarily included trials that investigated polatuzumab vedotin in the relapsed/refractory setting, which reflects the majority of available evidence at the time of analysis. We agree that the treatment context may influence efficacy and safety outcomes. The current data do not allow for a definitive conclusion regarding first-line use, as only a limited number of studies (such as the POLARIX trial) have evaluated polatuzumab vedotin in previously untreated patients. Subgroup analyses were not feasible due to the small number of first-line studies included. Therefore, our conclusions are most applicable to relapsed/refractory NHL, and we have clarified this in the revised manuscript. We recommend that future studies explore comparative outcomes in both treatment lines to inform more tailored conclusions.
- On the same line, table 1 should detail more precisely the clinical context of the trials (first line? Relapsed/refractory?)
Response: All included trials investigated polatuzumab vedotin in the relapsed/refractory setting. So, there is no study dealing with first-line treatment by polatuzumab vedotin for non-Hodgkin lymphoma (NHL) patients.
- The last sentence of page 5, “All included studies employed…” may be somewhat confusing and should be reformulated. Is “pinatuzumab vedotin” correct?
Response: Yes, it is correct. These trials were comparing polatuzumab vedotin with pinatuzumab vedotin.
- Could the authors define whether the rate of adverse events listed in their systematic review was dependent upon age? Or upon the specific treatment regimen that was utilized?
Response: Thank you for your valuable observation. Regarding the relationship between adverse event rates and age, we note that the populations across all included studies in our systematic review predominantly consisted of older adults, with a reported age range of 62 to 75 years. Due to this narrow age distribution, we were unable to conduct meaningful subgroup analyses to determine whether adverse event rates varied significantly by age. As for the specific treatment regimens, some variability in adverse event profiles was observed across different combinations involving polatuzumab vedotin (e.g., Pola-BR vs. Pola-R-CHP). However, due to heterogeneity in reporting formats and outcome definitions across studies, and the limited number of trials using the same comparator regimens, a robust regimen-specific meta-analysis of adverse events was not feasible.
- In view of Precision Medicine, one important issue is trying to understand which patients benefit the most from polatuzumab vedotin based on the molecular characteristics of the disease. Importantly, liquid biopsy of circulating tumor DNA has been shown to identify patients with optimal responses to standard treatment with R-CHOP, as recently reviewed and updated in detail (Almasri et al., Int J Mol Sci. 2025, 19;26(10):4869. doi: 10.3390/ijms26104869). In their Discussion, the authors should mention the role of liquid biopsy in lymphoma and state that an important future perspective will be to apply liquid biopsy to studies using polatuzumab vedotin in order to facilitate the early identification of responders versus non-responders (e.g. Moia et al. Blood Adv. 2025, 8;9(7):1692-1701. doi: 10.1182/bloodadvances.2024014136).
Response: Thank you for your valuable addition related to liquid biopsy. We have added a paragraph in the discussion related to liquid biopsy for B and T cell lymphomas.
Reviewer 2 Report
Comments and Suggestions for Authors
The authors present a systematic review and meta-analysis evaluating the efficacy of Polatuzumab Vedotin, an antibody-drug conjugate targeting CD79B, in the treatment of Non-Hodgkin Lymphoma (NHL). This is a clinically relevant and timely topic, as Polatuzumab Vedotin has been gaining traction in the treatment of relapsed or refractory diffuse large B-cell lymphoma (DLBCL) and follicular lymphoma. The authors aim to aggregate current evidence to determine overall treatment efficacy.
Recommendations:
The title is somewhat redundant. Consider revising to:
“Efficacy of Polatuzumab Vedotin Targeting CD79B in the Treatment of Non-Hodgkin Lymphoma: A Systematic Review and Meta-analysis”
Figure 3 resolution is low.
Author Response
R2
Comments and Suggestions for Authors
The authors present a systematic review and meta-analysis evaluating the efficacy of Polatuzumab Vedotin, an antibody-drug conjugate targeting CD79B, in the treatment of Non-Hodgkin Lymphoma (NHL). This is a clinically relevant and timely topic, as Polatuzumab Vedotin has been gaining traction in the treatment of relapsed or refractory diffuse large B-cell lymphoma (DLBCL) and follicular lymphoma. The authors aim to aggregate current evidence to determine overall treatment efficacy.
Recommendations:
The title is somewhat redundant. Consider revising to:
“Efficacy of Polatuzumab Vedotin Targeting CD79B in the Treatment of Non-Hodgkin Lymphoma: A Systematic Review and Meta-analysis”
Figure 3 resolution is low.
Response: Dear Reviewer, we are glad you found our manuscript interesting. We appreciate your suggestion about the title. We have updated the title to improve its credibility and readability for readers.
Also, we tried to improve the quality of all the Images
Reviewer 3 Report
Comments and Suggestions for Authors
The topic of Polatuzumab Vedotin as a targeted therapy for Non-Hodgkin Lymphoma (NHL) is both relevant and timely, given the increasing incidence of NHL and the need for effective treatments for relapsed/refractory cases.
The research design follows the PRISMA guidelines, and the use of a systematic review and meta-analysis is appropriate.
1) The results are presented clearly, with appropriate statistical analyses. However, some results, especially concerning overall survival (OS), lack a thorough exploration of potential confounders. The discussion would benefit from a more detailed examination of why OS did not show significant improvement, including considerations of patient demographics, treatment histories, and the biological characteristics of NHL subtypes.
2) In addition, data recently presented at EHA should be included, even if still in form of abstracts
3 )The introduction and discussion, can be made concise to improve readability.
Author Response
R3
Comments and Suggestions for Authors
The topic of Polatuzumab Vedotin as a targeted therapy for Non-Hodgkin Lymphoma (NHL) is both relevant and timely, given the increasing incidence of NHL and the need for effective treatments for relapsed/refractory cases.
The research design follows the PRISMA guidelines, and the use of a systematic review and meta-analysis is appropriate.
1) The results are presented clearly, with appropriate statistical analyses. However, some results, especially concerning overall survival (OS), lack a thorough exploration of potential confounders. The discussion would benefit from a more detailed examination of why OS did not show significant improvement, including considerations of patient demographics, treatment histories, and the biological characteristics of NHL subtypes.
Response: Dear Reviewer, Thank you for your thoughtful feedback. We appreciate your suggestion regarding the need for a deeper exploration of the overall survival (OS) results. We have revised the Discussion section to include a more detailed analysis of potential factors that may have influenced the lack of significant OS improvement.
2) In addition, data recently presented at EHA should be included, even if still in form of abstracts
Response: Based on our inclusion and exclusion criteria, we have added all relevant data.
3 ) The introduction and discussion can be made concise to improve readability.
Response: We have revised and improved the introduction and discussion to the best of our knowledge.
Round 2
Reviewer 1 Report
Comments and Suggestions for Authors
The authors have adequately addressed the issues that had been raised. No further comments from my side.
Reviewer 3 Report
Comments and Suggestions for Authors
no further comment